# Towards Understanding PRPS1 as a Molecular Player in Immune Response in Yellow Drum (*Nibea albiflora*)

**DOI:** 10.3390/ijms23126475

**Published:** 2022-06-09

**Authors:** Qianqian Tian, Wanbo Li, Jiacheng Li, Yao Xiao, Baolan Wu, Zhiyong Wang, Fang Han

**Affiliations:** 1Key Laboratory of Healthy Mariculture for the East China Sea, Ministry of Agriculture and Rural Affairs, Fisheries College, Jimei University, Xiamen 361021, China; tianqianqian0511@163.com (Q.T.); li.wanbo@jmu.edu.cn (W.L.); 202011908012@jmu.edu.cn (J.L.); 202012951035@jmu.edu.cn (Y.X.); wbl5160691422@163.com (B.W.); zywang@jmu.edu.cn (Z.W.); 2Laboratory for Marine Fisheries Science and Food Production Processes, Qingdao National Laboratory for Marine Science and Technology, Qingdao 266237, China

**Keywords:** innate immunity, PRPS1, GST pull-down, immunohistochemistry, Myd88, *Vibrio harveyi*, *Nibea albiflora*

## Abstract

Phosphoribosyl pyrophosphate synthetases (EC 2.7.6.1) are key enzymes in the biological synthesis of phosphoribosyl pyrophosphate and are involved in diverse developmental processes. In our previous study, the *PRPS1* gene was discovered as a key disease-resistance candidate gene in yellow drum, *Nibea albiflora*, in response to the infection of *Vibrio harveyi*, through genome-wide association analysis. This study mainly focused on the characteristics and its roles in immune responses of the *PRPS1* gene in yellow drum. In the present study, the *NaPRPS1* gene was cloned from yellow drum, encoding a protein of 320 amino acids. Bioinformatic analysis showed that *Na*PRPS1 was highly conserved during evolution. Quantitative RT-PCR demonstrated that *NaPRPS1* was highly expressed in the head-kidney and brain, and its transcription and translation were significantly activated by *V. harveyi* infection examined by RT-qPCR and immunohistochemistry analysis, respectively. Subcellular localization revealed that *Na*PRPS1 was localized in cytoplasm. In addition, semi-in vivo pull-down assay coupled with mass spectrometry identified myeloid differentiation factor 88 (MyD88) as an *Na*PRPS1-interacting patterner, and their interaction was further supported by reciprocal pull-down assay and co-immunoprecipitation. The inducible expression of *MyD88* by *V. harveyi* suggested that the linker molecule MyD88 in innate immune response may play together with *Na*PRPS1 to coordinate the immune signaling in yellow drum in response to the pathogenic infection. We provide new insights into important functions of PRPS1, especially PRPS1 in the innate immunity of teleost fishes, which will benefit the development of marine fish aquaculture.

## 1. Introduction

Yellow drum (*Nibea albiflora*) belongs to the Sciaenidae family, and it is mainly distributed from the South China Sea to the coastal waters of Japan and Korea [1,2]. It is one of the main fishery products and is an emerging farmed fish with substantially high economic and breeding values [1,3]. The yellow drum aquaculture is increasingly gaining popularity in the coastal regions of southeast China; however, it is accompanied by threats from bacterial and viral infection. Red-head disease caused by *Vibrio harveyi* is the most harmful disease threatening aquacultured yellow drum, which is mainly manifested as erythema or even ulcer on the head, and atrophy of anatomical viscera. It has a high infection rate in juvenile fish, and the mortality rate can exceed 50% and reach even close to 100%, causing great economic losses [1,2,3,4,5]. To uncover the molecular mechanism governing the immune response of yellow drum against the infection of *V. harveyi*, a genome-wide association analysis was performed and a set of resistance trait-linked SNPs (single nucleotide polymorphism) have been identified in our previous study [5]. Through datamining and functional genomics analysis, a gene encoding phosphoribosyl pyrophosphate synthetase 1 (PRPS1) was isolated as a putative disease resistance gene against *V. harveyi* in yellow drum.

Phosphoribosyl pyrophosphate synthetase (PRPS; EC 2.7.6.1), also known as ribose-phosphate pyrophosphokinase, is a key enzyme in nucleotide metabolism, catalyzing ribo-5-phosphate (R5P) and ATP to phosphoribosyl pyrophosphate (PRPP) and AMP. PRPP is an important metabolic intermediate involved in de novo and remedial synthesis of purine and pyrimidine nucleotides, providing key structural components of certain nucleotide coenzymes such as coenzyme I and coenzyme II, and amino acids such as histidine and tryptophan [6,7,8,9,10]. The enzyme unit of PRPS is a hexamer, consisting of three homologous dimers arranged in a propeller shape [9], with an active site and two regulatory allosteric sites in each homodimer.

In human, three isoforms of PRPS have been identified, including PRPS1, PRPS2, and PRPS1L1 (PRPS3). PRPS1 plays an important role in controlling tumorigenesis, promoting cell apoptosis, and inhibiting cell proliferation, rendering PRPS1 a potential target for treatment of tumors [9,10]. Abnormal activity of this gene has some clinical consequences. For example, PRPS1 hyperactivity as a result of elevated PRPS1 transcription or gain-of-function mutations can lead to the classic X-linked disorder associated with hyperuricemia and hyperuricosuria, often causing gout in females and children [11]. In contrast, significant decrease in transcriptional level of PRPS1 was observed in human Hutchinson-Gilford progeria syndrome (HGPS)-derived fibroblast cell lines [12]. Mildly compromised PRPS1 activity caused by loss-of-function mutations leads to X-linked non-syndromic sensorineural hearing loss (DFNX-2) [13], and severe loss of PRPS1 activity in a missense mutation is often represented by the more severe Arts syndrome, in which patients present with a “triad” of preverbal hearing loss, early-onset optic atrophy, and severe sensory neuromotor neuropathy [14]. PRPS is rarely studied in fish, PRPS polymerization influences lens fiber organization in zebrafish, and the loss of PRPS activity regulates retinal development in zebrafish [15]. There is no report on the research of PRPS in fish immunity.

The post-translational modification of PRPS1 protein can regulate its activity. It was demonstrated that AMPK (AMP-activated protein kinase)-mediated phosphorylation of PRPS1 S180 residue leads to conversion of PRPS hexamer to monomer and thereby inhibits PRPS1 activity [9]. On the other hand, point mutation of PRPS1 at S180 to non-phosphorylatable alanine residue offered uninhibited activity of PRPS1 [9]. In contrast, the phosphorylation of PRPS1 at S103 mediated by CDK1 (cyclin dependent kinase 1) demonstrated enhanced enzyme activity, and the phosphorylation at T228 of human PRPS1 imparted boosted catalytic activity to repair DNA damage induced by ionized radiation [8]. The diverse effects of phosphorylation on PRPS1 activity imply the complex tunability of PRPS1 activity by phosphorylation.

In the present study, the PRPS1 from *Nibea albiflora* (*NaPRPS1*) was cloned and characterized. The tissue expression pattern and subcellular localization of *Na*PRPS1 were investigated. The transcript level and protein level of *Na*PRPS1 in yellow drum were determined by RT-qPCR and immunohistochemistry after the challenge with *V. harveyi*, respectively. In addition, pull-down assay and mass spectrometry were employed to search for the target proteins interacting with *Na*PRPS1, and their biological relevance in immune response was discussed. This is the first time a study has identified this gene in marine fish; therefore, this gene’s function in the innate immunity of teleost fishes was studied for the first time.

## 2. Results and Discussion

### 2.1. Molecular Characteristics of NaPRPS1

The genomic DNA sequence of *NaPRPS1* in *N. albiflora* was 2691 bp in length, containing 7 exons and 6 introns (Figure 1A). The 6 introns were 495, 641, 162, 165, 134, and 131 bp in length, respectively. All introns conformed to the typical intron-splicing motif (GT/AG rule). The encoded protein of *NaPRPS1* by the coding region of 963 bp consisted of 320 amino acids, with residues of serine (Ser), threonine (Thr), and tyrosine (Tyr) (Figure 1B), providing putative phosphorylation-targeting sites for its activity regulation. The alignment of *Na*PRPS1 full-length protein sequence against the yellow drum proteome database did not reveal the existence of other isoforms, suggesting *NaPRPS1* is a single copy gene in the yellow drum genome. Protein sequence analysis indicated that there was no signal peptide or transmembrane domain in *Na*PRPS1. The deduced molecular weight was 34.6 kDa, and the calculated isoelectric point was 6.31. Conserved domain search showed the presence of typical phosphoribose pyrophosphate synthase superfamily specific binding site domains (Figure 1B). The three-dimensional structure of *Na*PRPS1 was computed, showing the spatial arrangement of secondary elements including α-helices and β-sheets, with a 96.07% similarity with the crystal structure of human PRPS1 (PDB: 2h06.1.B) (Figure 1C). Sequence alignment showed that fish and mammal PRPS1 proteins were highly evolutionarily conserved, with *Na*PRPS1 sharing 96.85% and 86.54% identity with its homologs in large yellow croaker (*Larimichthys crocea*) and in humans (Homo sapiens), respectively (Figure 2A). The phylogenetic analysis indicated that *Na*PRPS1 shared the closest relationship with the homolog in large yellow croaker (Figure 2B).

### 2.2. NaPRPS1 Expression Tissue Specificity and Subcellular Localization

Quantitative reverse transcription PCR was employed to investigate the expression of *Na*PRPS1 in different tissues of yellow drum. The normalized expression profile indicated that *NaPRPS1* is highly expressed in the head-kidney and brain, and its transcripts can be readily detected in other tissues including heart, liver, spleen, grill, and intestine (Figure 3A). The highest expression of *NaPRPS1* in the head-kidney is coincident with the important role of the head-kidney in immunity and is consistent with the observations of the most abundant transcripts of *PRPS1* in the adrenal gland and brain of rats and humans [16,17,18].

To gain an insight into the subcellular localization of *Na*PRPS1 protein, an expression cassette was constructed to express a GFP tagged *Na*PRPS1 fusion protein. Fluorescent microscopy of the transfected HEK 293T cells harboring GFP-*Na*PRPS1 indicated that *Na*PRPS1 protein was localized in the cytoplasm, while the free GFP protein was found both in the cytoplasm and nucleus (Figure 3B). The observation of the cytosolic localization of *Na*PRPS1 is partially supported by its lack of specified signals for nuclear localization or plasma membrane localization and is consistent with the predicted cytosolic localization of human PRPS1 by GeneCard database.

### 2.3. Defence Response of NaPRPS1 against V. harveyi Infection

Given the close relationship of *NaPRPS1* to the resistance trait of yellow drum to *V. harveyi*, the expression of *NaPRPS1* over a time course in response to the attack of *V. harveyi* was investigated. The transcript level of *NaPRPS1* was substantially increased in head-kidney, liver, and brain as early as 6 h after infection, and reached a peak at 72 h post infection, with 4.25 times, 4.18 times, and 3.53 times higher expression level than the control, respectively (Figure 4A). There was no significant change in the expression in spleen across the time course tested. The high expression of *NaPRPS1* in the brain of yellow drum may be symptomatic of the “red head disease” that accompanies *V. harveyi* infection.

Based on the detection of a significantly upregulated expression of *NaPRPS1* in response to the bacterial infection by RT-qPCR, immunohistochemical staining was further developed to examine the *Na*PRPS1 protein distribution in yellow drum tissues after the pathogenic attack. The head-kidney and brain tissues from the resistant fish were prepared and probed with anti-*Na*PRPS1 antibody. The immunohistochemistry analysis indicated markedly stronger expression of *Na*PRPS1 in both head-kidney and brain tissues in the pathogen-stimulated fish, compared to the sporadic signal detection in the unchallenged control fish (Figure 4B). Taken together, the activated expression of *NaPRPS1* against *V. harveyi* infection at both transcriptional level and translational level suggested a positive role of *Na*PRPS1 in response to the pathogen.

### 2.4. NaPRPS1-Interacting Protein Isolation and Identification

To obtain a better understanding of the regulatory role of *Na*PRPS1 in defense response, a semi-in vivo pull-down assay was performed, aiming to isolate its interacting partners. To fulfil this purpose, the GST-*Na*PRPS1 fusion protein was expressed in *E. coli* BL21 (DE3) and purified by affinity chromatography (Figure 5A), and the recombinant protein was used to probe the proteins physically interacting with *Na*PRPS1 in yellow drum tissues. One of the differential protein signals detected specifically in the pull-down by GST-*Na*PRPS1 (Figure 5B) was identified by mass spectrometry to be MyD88 (myeloid differentiating factor 88) (Figure 5C), which is normally considered as an adaptor mediating signal transduction in immune response.

### 2.5. Validation of NaPRPS1-MyD88 Interaction

MyD88 is a cytoplasmic protein linking the perception of external stimuli by toll-like receptor (TLR) or interleukin-1 (IL1) receptor and the downstream signal cascade involving phosphorylation events and ultimate immune responses [19,20,21]. The subcellular localization of *Na*PRPS1 in the cytoplasm provides the possibility of its physical interaction with MyD88. Based on the primary identification of MyD88 as an interacting molecule with *Na*PRPS1 through the semi-in vivo screening and mass spectrometry analysis, the interactions between them were further validated by the reciprocal pull-down assay using GST-Myd88 as the bait and co-immunoprecipitation assay. The recombinant protein GST-Myd88 was expressed in *E. coli* BL21 (DE3) (Figure 6A) and demonstrated its ability to specifically pull down *Na*PRPS1 from yellow drum cell lysate detected, which can be detected by a western blot (Figure 6B). Furthermore, when the *Na*PRPS1 protein was immunoprecipitated by anti-*Na*PRPS1 antibody, the MyD88 protein was co-immunoprecipitated specifically together with *Na*PRPS1 in the protein extract of transfected HEK 293T cells (Figure 6C), suggesting the formation of *Na*PRPS1-MyD88 complex in vivo. In mammals, most TLR molecules can interact with MyD88 or its analogue (MyD88 adaptor like, Mal) through the TIR (Toll/interleukin-1 receptor) domain of these adaptors to transmit signals and activate immune response [22,23]. After the formation of MyD88-TLR/IL-1R complex, the recruited kinase can phosphorylate and activate the downstream signaling molecules. It is conceivable that the interaction between *Na*PRPS1 and MyD88 in yellow drum may lead to phosphorylation of *Na*PRPS1 and thus activate its activity.

### 2.6. Participation of MyD88 in Defence Response

To further support the participation of *MyD88* in the defense response of yellow drum to *V. harveyi* infection, the expression pattern of *MyD88* after the bacterial challenge was investigated. The basal expression of *MyD88* in different tissues was monitored through RT-qPCR, showing that it was highly expressed in spleen, head-kidney, gill, and brain (Figure 7A). After the infection by *V. harveyi*, the transcript level of *MyD88* was significantly elevated in liver, spleen, head-kidney, and brain at 48 h after the stimulation (Figure 7B). Early response of *MyD88* to the pathogen attack was detected at 6 h post infection in the tissues of liver, spleen, and brain (Figure 7B). The induced expression of *MyD88* was also documented in half-smooth tongue sole after bacterial inoculation [23] and in Japanese flounder after immunization with PAMP such as LPS, Poly I:C, and peptidoglycan [24]. The activated expression of *MyD88* in response to the pathogenic stimulus [25,26,27,28,29], in concert with the transcriptional elevation of *NaPRPS1*, provided additional evidence to substantiate that *Na*PRPS1 and MyD88 may cooperate to regulate the immune response of yellow drum.

## 3. Materials and Methods

### 3.1. Experimental Fish and Immune Challenge

Healthy juvenile yellow drum (fish length 3.28 ± 1.71 cm) were obtained from a mariculture farm in Ningde, Fujian, China. The cultivation and infection of yellow drum with *V. harveyi* were conducted according to a previously established method [5]. Different tissues including muscle, head-kidney, skin, intestine, gill, spleen, liver, heart, and brain were obtained from six healthy *N. albiflora* to examine the spatial expression pattern of *Na*PRPS1. In addition, samples of head-kidney, liver, spleen, and brain of *N. albiflora* were collected at 0 h, 6 h, 12 h, 24 h, 48 h, 72 h, and 96 h after inoculation with *V. harveyi* to study the temporal expression of *NaPRPS1* in response to the bacterial challenge. The collected tissues were immediately preserved in liquid nitrogen and then stored in −80 °C freezer until use. The survivals of yellow drum at 14 days after the pathogenic attack were marked as resistant fish and their head-kidney and brain tissues were collected and placed in 4% paraformaldehyde at room temperature for immunohistochemistry. The samples collected from unchallenged fish were included as control.

### 3.2. RNA Extraction and cDNA Synthesis

Total RNA was isolated from the above samples using TransZol Up Plus RNA Kit (TransGen Biotech, Shaihai, China) according to the manufacturer’s instruction. First strand cDNA was synthesized from total RNA using GoScrip Reverse Transcription System (Promega, Madison, Wisconsin, USA) in accordance with the manufacturer’s protocol.

### 3.3. Cloning of NaPRPS1 and MyD88

The open reading frame (ORF) of *NaPRPS1* was screened from the transcriptome of *N. albiflora* generated in our laboratory. A pair of gene specific primers (*PRPS1-cF* and *PRPS1-cR*) with *Eco*R I and *Xho* I restriction sites (Table 1) were designed according to the manual of ClonExpress II One Step Cloning Kit (Vazyme Biotech, Nanjing, China) to amplify the coding region of *NaPRPS1*, and the PCR product was directly cloned into the linearized pGEX-6P-1 vector to make an in-frame fusion with the glutathione S-transferase (GST) tag. Similarly, the coding region of *MyD88* was amplified with primers *MyD88-cF* and *MyD88-cR* (Table 1) and cloned into pGEX-6P-1 vector. The resulting constructs, namely GST-*Na*PRPS1 and GST-MyD88, were sequenced by BioSune Biotech (Shanghai, China) to verify the sequence fidelity.

### 3.4. NaPRPS1 Protein Sequence Analysis

The molecular weight, theoretical pI, and amino acid composition of the *Na*PRPS1 protein were investigated through the online ProtParam tool (www.expasy.org 5 June 2021). The protein domain detection was performed using the web-based SMART tool (smart.embl-heidlberg.de 5 June 2021). Phosphorylation sites were predicted using the online NetPhos 3.1 server (services.healthtech.dtu.dk 6 June 2021). The prediction of signal peptide was conducted with SignalP program (services.healthtech.dtu.dk 7 June 2021). The sequence alignment of *Na*PRPS1 and its homologs was conducted with Clustal Omega program (www.ebi.ac.uk 8 June 2021), and the result was presented with the online sequence manipulation suite (www.detaibio.com/sms2 9 June 2021). The tertiary structure of *Na*PRPS1 was predicted with SWISS-MODEL workspace (swissmodel.expasy.org 10 June 2021) and visualized by Visual Molecular Dynamics 1.9.4a51. A phylogenetic tree was constructed through the neighbor-joining (N-J) method in MEGA 11.

### 3.5. Real-Time PCR Analysis

Quantitative reverse transcription PCR (RT-qPCR) was performed to determine the tissue-specific expression of *NaPRPS1* and *MyD88* in healthy fish, as well as their time-course expression in response to a *V. harveyi* challenge. Gene-specific primers for *NaPRPS1* (*PRPS1**-qF* and *PRPS1-qR*) and *MyD88* (*MyD88**-qF* and *MyD88**-qR*) were used to amplify the *NaPRPS1* and *MyD88* fragments, respectively, and *β-actin* was selected as a reference gene amplified with primers *β-actin**-qF* and *β-actin-qR* (Table 1) to normalize their expression levels. The RT-qPCR using ChamQ Universal SYBR qPCR Master Mix (Vazyme Biotech, Najing, China) was performed on an Applied Biosystems QuantStudio 6 & 7 Real-time PCR System (Application Biosystems, Foster, USA). All reactions were run in triplicate. The relative transcript levels were quantified on a relative scale by the 2^−ΔΔCT^ method.

### 3.6. NaPRPS1 Subcellular Localization

To study the subcellular localization of the *Na*PRPS1 protein, the coding region of *Na*PRPS1 was amplified with specific primers, *PRPS1-sF* and *PRPS1-sR* (Table 1), and recombined into the pEGFP-N1 vector with ClonExpress II One Step Cloning Kit (Vazyme Biotech, Nanjing, China). The recombinant was designated as GFP-*Na*PRPS1. Human embryonic kidney 293T (HEK 293T) cells were seeded into 6-well plates and cultured for 24 h in DMEM with 10% fetal bovine serum and 1% penicillin-streptomycin solution (Biological Industries) under a humidified condition with 5% CO_2_ at 37 °C [14]. Afterwards, 1 μg of plasmids of GFP-*Na*PRPS1 and pEGFP-N1 (control) were respectively transfected into the cells with Lipo 8000 Transfection Reagent (Beyotime, Shanghai, China). The protein subcellular localization was observed with the confocal fluorescence microscopy Leica TCS SP8 system (Leica, Heidelberg, Germany).

### 3.7. Expression and Purification of Recombinant Protein

The constructed expression plasmid containing GST-*Na*PRPS1 or GST-MyD88 was transformed into *E. coli* BL21 (DE3) for prokaryotic expression. The transformed cells were subcultured and isopropyl β-D-thiogalactoside (IPTG) with the final concentration of 0.01 mM was added when the OD600 value of the subculture reached 0.6–0.8 to induce the expression of the recombinant protein at 16 °C with shaking at 200 rpm for 20 h.

Protein purification was carried out with reference to the method described previously [30]. Briefly, the induced cells were harvested by spinning at 4000× *g* for 15 min at 4 °C, and then resuspended in ice-cold sodium phosphate buffer (PBS, pH 7.3) and sonicated for 10 min on ice. The supernatant of the cell lysate after sonication was incubated with glutathione-agarose beads (Sigma, St. Louis, Mo, USA) at 4 °C for 2 h with gentle rotation. The beads were washed 3 times with ice-cold PBS, and then incubated in reducing buffer (50 mM Tris-HCl, 10 mM reduced glutathione, pH 8.0) at room temperature for 10 min. After centrifugation at 1000× *g* for 5 min, the supernatant was collected and analyzed by SDS-PAGE. The purified protein was saved at −80 °C until use.

### 3.8. Preparation of Antibody and Purification

To prepare the antibody against *Na*PRPS1, the purified recombinant GST-*Na*PRPS1 was digested with PreScission Protease (Beyotime, Beijing, China) on a column to release the tag-free *Na*PRPS1, and then the free *Na*PRPS1 protein was used to immunize mice by intradermal injection once a week over a 4-week period. The antigen of 5 μg of *Na*PRPS1 was mixed with an equal volume of Freund’s complete adjuvant (Sigma) for the first injection. Subsequent injections were conducted with 5 μg of antigen mixed with an equal volume of Freund’s incomplete adjuvant (Sigma). The immunoglobulin (IgG) fraction was purified by protein A-Sepharose (Bio-Rad, California, USA) and then stored at −80 °C. The titration and specificity of the antibody generated was tested in our laboratory by ELISA (enzyme-linked immunosorbent assay).

### 3.9. Immunohistochemistry

Immunohistochemical staining to examine *Na*PRPS1 expression was adapted from the method described previously [31]. The collected head-kidney and brain tissues from the healthy fish (not challenged) and the resistant fish (the survivals at 14 days after challenge with *V. harveyi*) were fixed with 4% paraformaldehyde, embedded in paraffin, and sectioned through routine procedures. After removing the paraffin and dehydrating agents, the sections were incubated with 3% H_2_O_2_ in methanol for 25 min to inactivate the endogenous peroxidase activity and blocked in 3% BSA-PBS. The anti-*Na*PRPS1 mouse antibody (1:200) was added to the sections and incubated overnight at 4 °C. After 3 washes with PBS, the sections were incubated with secondary antibody (1:5000 dilution of HRP-labeled goat anti-mouse antibody) for 40 min prior to staining with DAB and hematoxylin. The processed sections were examined under light microscope.

### 3.10. GST Pull-Down Assay and Mass Spectral Analysis

The GST pull-down assay was performed according to the method described [32]. The purified GST and GST-*Na*PRPS1 proteins were incubated with glutathione beads with agitation for 2 h at 4 °C to be used as bait. To prepare the input total protein, 100 mg of pooled yellow drum tissues from brain, head-kidney, liver, blood, spleen, and heart were homogenized in 1 mL of GST binding buffer (200 mM NaCl, 20 mM Tris-HCl, 1 mM EDTA, PMSF 2 mM, pH 7.4) at 4 °C, followed by sonication on ice for 40 s, and then 0.5% NP-40 and 1% TritonX-100 were added into the homogenate. After gentle agitation at room temperature for 1.5 h, the homogenate was centrifuged at 10,000× *g* for 15 min at 4 °C, and the supernatant was incubated with the immobilized GST-*Na*PRPS1 and GST on glutathione beads for 4 h at 4 °C. The beads were washed with GST binding buffer 5 times and resolved on 12% SDS-PAGE gel. The gel was developed with silver staining and putative *Na*PRPS1-interacting proteins were recovered from the gel for mass spectrometry. Briefly, protein bands were excised from the polyacrylamide gel and subjected to in-gel digestion by trypsin. The mass spectra of the peptides were obtained with a time-of-flight delayed extraction MALDI MS (Bruker Autoflex, Shanghai, China). A nitrogen laser (337 nm) was used to irradiate the sample. Spectra were acquired in reflectron mode in the mass range of 600–3500 Da. A near point calibration was applied and a mass tolerance of 100 ppm used. Data mining was performed with the Mascot search engine against the GenBank database.

### 3.11. Reverse GST Pull-Down and Co-Immunoprecipitation Analysis

The reciprocal pull-down assay with immobilized GST-MyD88 was performed as described above to verify the interaction between *Na*PRPS1 and MyD88, and the membrane was probed with an anti-*Na*PRPS1 antibody to detect the pull-down by MyD88 and GST control.

For co-immunoprecipitation (Co-IP) assay, *Na*PRPS1 and MyD88 were amplified with primer pairs of *PRPS1-eF/-eR* and *MyD88-eF/-eR* (Table 1), respectively, and cloned into pcDNA3.1(+)/Myc-His B vector. To be used as a control, β-Actin was also obtained by PCR with primers *β-actin-eF/-eR* and cloned into pcDNA3.1(+)/Myc-His B vector. The constructs alone or the defined combinations (1 μg of each plasmid) were introduced into HEK 293T cells with Lipo 8000 Transfection Reagent. At 36 h post transfection, trypsinization was performed for 3 min, and the cells were collected by centrifugation at 1000 rpm for 3 min. Cell lysis buffer (50 mM Tris pH7.4, 150 mM NaCl, 1% NP-40, 1% SDS, PMSF) was used to lyse the cells in an ice water bath for 30 min with intermittent mixing every 5 min. After centrifugation at 13,000 rpm for 10 min at 4 °C, the supernatant was transferred to a pre-cooled 1.5 mL centrifuge tube to be used as the total cell protein extract for the following immunoprecipitation assay.

The anti-*Na*PRPS1 antibody was immobilized on protein A resins and incubated with the above cell protein extract at 4 °C for 2 h, and then loaded to a column. After the extract ran through the column by gravity flow, the beads were washed 3 times with the cell lysis buffer. The beads were boiled, and the proteins were resolved by SDS-PAGE and subjected to western blot analysis with an anti-Myc antibody.

## 4. Conclusions

In summary, the *NaPRPS1* gene serving as a monocopy in yellow drum was assumed, based on our previous genome-wide association analysis, to be closely linked with a disease resistance trait. The expression of *NaPRPS1* was inducible at both transcriptional and translational levels in response to the infection of *V. harveyi*. In addition, the MyD88 adaptor protein known in innate immune signaling was demonstrated to physically interact with *Na*PRPS1, orchestrating the defense response. The interaction between MyD88 and *Na*PRPS1 may facilitate the phosphorylation of *Na*PRPS1 to modulate its activity. However, we did not rule out the possibility of the regulation of MyD88 activity by *Na*PRPS1 or other unknown events. Our data are still preliminary, and further efforts will be implemented to better interpret the underlying mechanism for their interaction and their roles in immune responses in yellow drum.

## Figures and Tables

**Figure 1 ijms-23-06475-f001:**
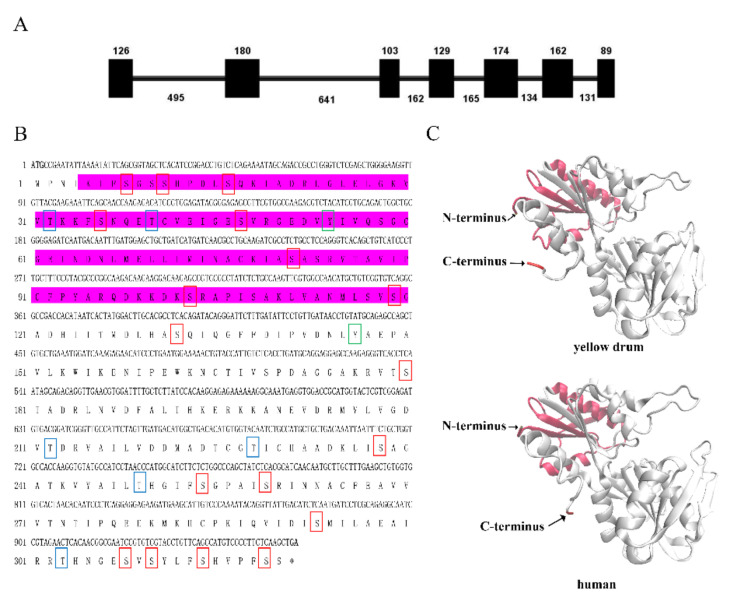
*NaPRPS1* gene structure and the three-dimensional structure of *Na*PRPS1 protein. (**A**) Genomic structure of *NaPRPS1*. Black boxes represent exons, and lines represent introns. The lengths of exons are shown on the top, and the lengths of introns are shown at the bottom; (**B**) The coding sequence of *NaRRPS1* and the deduced amino acid sequence. The nucleotides and amino acids are numbered along the right margin. The start and stop codons are in bold. Pribosyltran domain (Phosphoribosyl pyrophosphate synthase superfamily specific binding sites) is highlighted with pink. The predicted phosphyorylation sites on serine (S), threonine (T), and tyrosine (Y) residues are indicated by red, blue, and green boxes, respectively; (**C**) Structure analysis of *Na*RRPS1. Ribbon diagrams were shown for the three-dimensional structures of PRPS1 in yellow drum (left) and human (right). Spirals and arrows in the ribbon diagrams represent α-helix and β-sheet structures, respectively.

**Figure 2 ijms-23-06475-f002:**
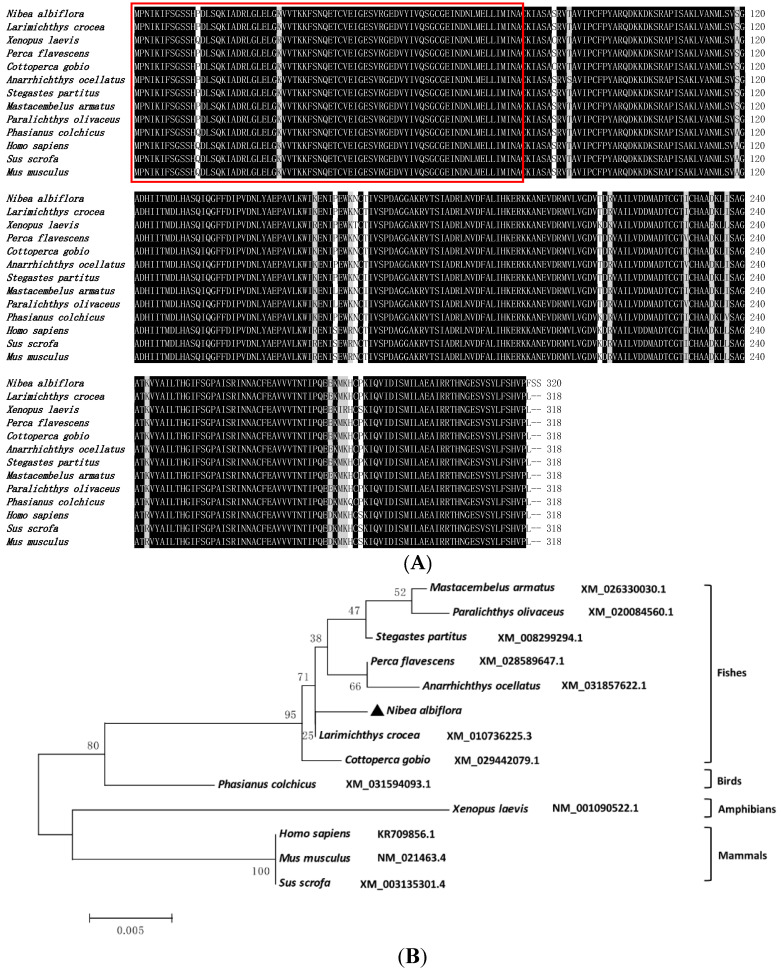
Sequence alignment and phylogenetic analysis of *Na*PRPS1 and its homologs. (**A**) Multiple sequence alignment of *Na*RRPS1 and its homologs. Pribosyltran domain is highlighted in red box; (**B**) Phylogenetic analysis of *Na*RRPS1 and its homologs. The protein sequences were defined with their respective Genbank accession number. The number at each node indicates the percentage of boot-strapping after 1000 replications. Scale bar refers to evolutionary distance of 0.05 million years.

**Figure 3 ijms-23-06475-f003:**
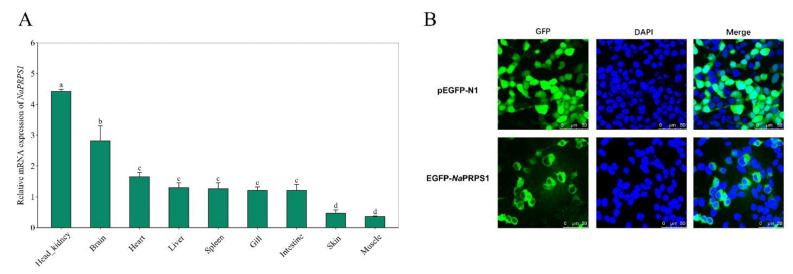
*NaPRPS1* tissue expression profile and the subcellular localization of *Na*PRPS1. (**A**) Relative transcript levels of *NaPRPS1* in different tissues. RT-qPCR was performed to determine the relative expression of *NaPRPS1* in different tissues of yellow drum, including muscle, head-kidney, skin, intestine, gill, spleen, liver, heart and brain. *β-actin* was used as an internal control to normalize the expression level. Data are mean ± SE (*n* = 6). The letters a, b, c and d denote statistical significance (*p* < 0.05); (**B**) Subcellular localization of *Na*PRPS1 protein in transfected HEK 293T cells. The images were captured under confocal fluorescence microscopy. Scale bar = 50 µm.

**Figure 4 ijms-23-06475-f004:**
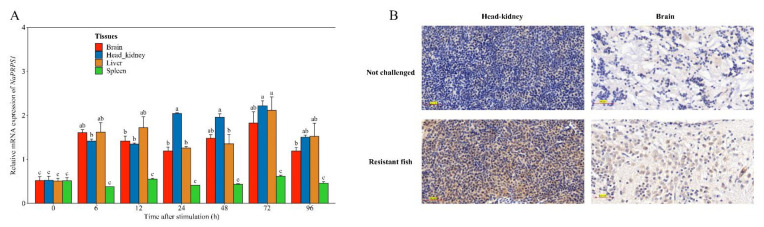
Defence response of *NaPRPS1* against *V. harveyi* infection. (**A**) Time-course expression of *NaPRPS1* in different tissues after *V. harveyi* infection. The relative expression levels of *NaPRPS1* were determined by RT-qPCR in spleen, head kidney, brain and liver at 0 h, 6 h, 12 h, 24 h, 48 h, 72 h and 96 h after *V. harveyi* challenge. *β-actin* was used as an internal control. Data are mean ± SE (n = 6). The letters a, b, c, represent statistical significance (*p* < 0.05); (**B**) Immunohistochemical staining of *Na*PRPS1 protein in head-kidney and brain of yellow drum. Positive signal was indicated with brown stain. Scale bar = 300 µm.

**Figure 5 ijms-23-06475-f005:**
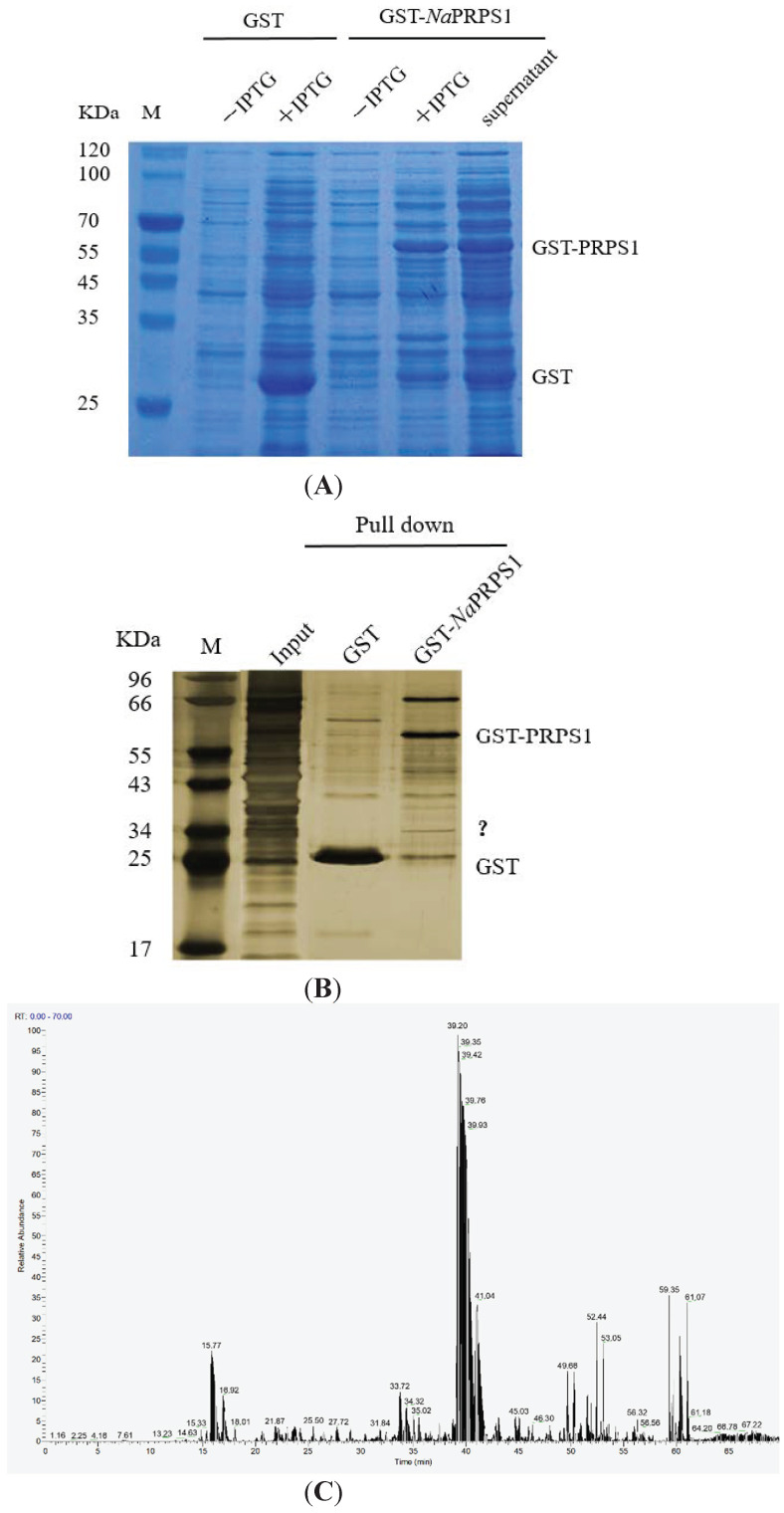
*Na*PRPS1-interacting protein isolation and identification. (**A**) SDS-PAGE analysis of the expression of recombinant *Na*PRPS1 in *E. coli*; (**B**) Semi-in vivo pull-down assay. GST or GST-*Na*PRPS1 was used as bait, and the total protein extract from yellow drum mixed tissue (brain, head-kidney, liver, blood, spleen, and heart) was used as input. The SDS-PAGE gel was developed with silver nitrate staining. Question mark indicates the differential signal detected by pull-down assay; (**C**) Protein identification by MALDI-TOF mass spectrometry.

**Figure 6 ijms-23-06475-f006:**
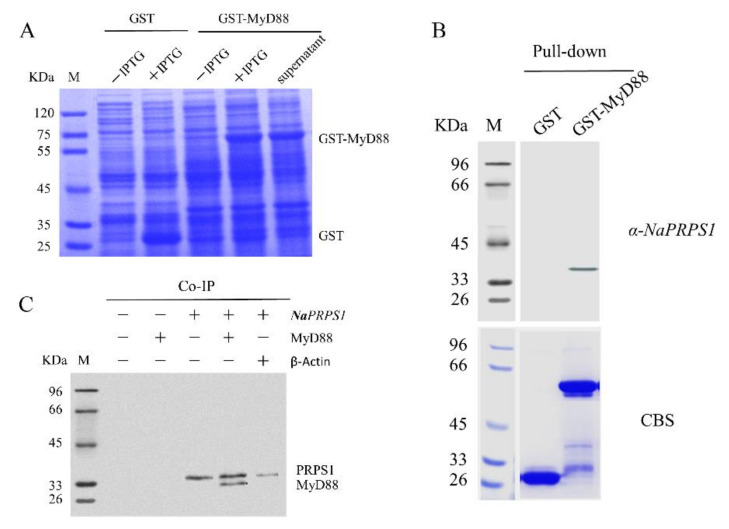
MyD88 expression and validation of *Na*PRPS1-MyD88 interaction. (**A**) SDS-PAGE analysis of the recombinant MyD88 expression in *E. coli*; (**B**) Semi-in vivo pull-down assay. The GST*-*MyD88 protein was immobilized and used as bait to pull down *Na*PRPS1 in the protein extract from yellow drum mixed tissue (brain, head-kidney, liver, blood, spleen, and heart). The bloting membrane was probed with anti-*Na*PRPS1 antibody for western blot assay (upper panel). The SDS-PAGE gel was visualized by coomassie blue staining (CBS) as loading control (bottom panel); (**C**) Co-IP assay. Anti-*Na*PRPS1 antibody was immobilized and used to precipitate *Na*PRPS1 protein, and the co-immunoprecipitated protein was detected by a western blot using an anti-Myc antibody.

**Figure 7 ijms-23-06475-f007:**
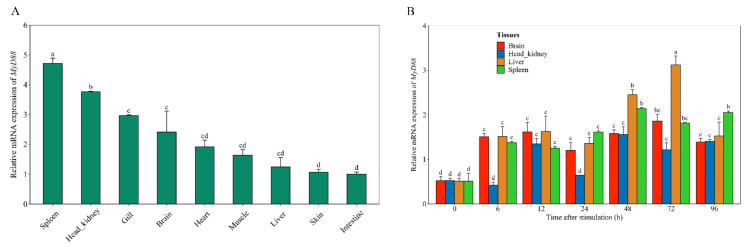
Tissue expression profile and Defence response of *MyD88* against *V. harveyi* infection. (**A**) RT-qPCR analysis of the relative expression of *MyD88* in different tissues of yellow drum. *β-actin* was used as an internal control. Data are mean ± SE (*n* = 6). The letters a, b, c, and d represent statistical significance (*p* < 0.05); (**B**) RT-qPCR analysis of the relative expression of *MyD88* in spleen, head kidney, brain, and liver after *V. harveyi* challenge at 0 h, 6 h, 12 h, 24 h, 48 h, 72 h, and 96 h. *β-actin* was used as an internal control. Data are mean ± SE (*n* = 6). The letters a, b, c, and d represent statistical significance (*p* < 0.05).

**Table 1 ijms-23-06475-t001:** Primers used for *NaPRPS1* and *MyD88* gene cloning and expression analysis.

Primer Name	Sequence (5′-3′)	Purpose
*PRPS1-cF*	cccctgggatccccg*GAATTC*ATGCCGAATATTAAAATATTC	Prokaryotic expression
*PRPS1-cR*	gtcacgatgcggccg*CTCGAG*TCAGCTTGAGAAGGGGACATG	
*MyD88-cF*	cccctgggatccccg*GAATTC*ATGGCGTGTTGCACAAGTC	
*MyD88-cR*	gtcacgatgcggccg*CTCGAG*CTTGGTCCTCTCATATGGC	
*PRPS1-sF*	ctaccggactcagat*CTCGAG*ATGCCGAATATTAAAATATTC	Subcellular localization
*PRPS1-sR*	gtsccgtcgactgca*GAATTC*CGGCTTGAGAAGGGGACATG	
*PRPS1-qF*	GCAAGACAAGAAGGACAAGAGCCGT	RT-qPCR analysis
*PRPS1-qR*	ATCAACAGGAATATCAAAGAATCCC	
*MyD88-qF*	ATGGCGTGTTGCGACAAGTCCGAGG	
*MyD88-qR*	TCCAGGGTGAGGCCGACCCTGTCTC	
*β-actin-qF*	TTATGAAGGCTATGCCCTGCC	
*β-actin-qR*	TGAAGGAGTAGCCACGCTCTGT	
*PRPS1-eF*	tgctggatatctgca*CTCGAG*ATGCCGAATATTAAAATATTC	Overexpression
*PRPS1-eR*	agtttttgttctaga*GAATTC*CGGCTTGAGAAGGGGACATG	
*MyD88-eF*	tgctggatatctgca*CTCGAG*ATGGCGTGTTGCACAAGTC	
*MyD88-eR*	agtttttgttctaga*GAATTC*CTTGGTCCTCTCATATGGC	
*β-actin-eF*	tgctggatatctgca*CTCGAG*ATGGAAGATGAAATCGCCGC	
*β-actin-eR*	agtttttgttctaga*GAATTC*GAAGCATTTGCGGTGGACG	

*Eco*R I (GAATTC) and *Xho* I (CTCGAG) enzyme restriction sites are italic.

## Data Availability

Not applicable.

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
