# Peer review of "Towards Understanding PRPS1 as a Molecular Player in Immune Response in Yellow Drum (Nibea albiflora)"

_ijms, 2022, doi:10.3390/ijms23126475_

Round 1
Reviewer 1 Report
The paper is a piece of nice work. The only suggestions I have is to eliminate Figure 2. At this stage of genomics, sequence comparisons and phylogenetic analysis of individual genes do not mean much.
Reviewer 2 Report
ijms-1743488 “Towards Understanding PRPS1 as a Molecular Player in Immune Response in Yellow Drum (Nibea albiflora)”.
GENERAL COMMENT:
The work entitled “Towards Understanding PRPS1 as a Molecular Player in Immune Response in Yellow Drum (Nibea albiflora)” is a good work; the subject is of current interest.
In this study, the PRPS1 from Nibea albiflora (NaPRPS1) was cloned and characterized in order to investigate the tissue expression specificity and subcellular localization.
The subject of the study is interesting.
Central argument is supported by evidence and analysis.
The methodology described by the author is accurate.
This work is a good work, and, in my opinion, it needs only minor changes; for this reason I require minor revision.
DETAILED COMMENT:
· Title
-The title is adequate.
· Abstract
In the abstract the objective of the study should be more clearly described.
· Introduction
I suggest expanding the section, it is not very exhaustive
· Results
This section is detailed and well written
· Discussion
The discussion section is adequately discussed and exhaustive.
· Tables and figures
Tables and Figures are clear and understandable.
· References
The references are adequate.
Author Response
Point 1: In the abstract the objective of the study should be more clearly described.
Response1: We thank the reviewer’s helpful suggestion. We have added a sentence “This study mainly focuses on the characteristics and its roles in immune responses of PRPS1 gene in yellow drum” in the abstract.
Point 2: Introduction: I suggest expanding the section, it is not very exhaustive.
Response 2: We thank the reviewer’s helpful suggestion. As suggested, the sentences have been add in third paragraph 3.Some others:
In fig 3A, fig 4A and fig7A&B, the results of RT-qPCR analysis of the relative expression of PRPS1 and MyD88, letters a, b, c and d which represent statistical significance (p < 0.05) have been reordered from high to bottom according to the drawing specification.